# Innovations in Snake Venom-Derived Therapeutics: A Systematic Review of Global Patents and Their Pharmacological Applications

**DOI:** 10.3390/toxins17030136

**Published:** 2025-03-14

**Authors:** Diana Carolina Zona Rubio, Diana Marcela Aragón, Izabel Almeida Alves

**Affiliations:** 1Grupo de Investigación Cuidado Cardiorrespiratorio, Universidad Manuela Beltrán, Bogotá 110231, Colombia; 2Departamento de Farmacia, Facultad de Ciencias, Universidad Nacional de Colombia, Bogotá 111321, Colombia; dmaragonn@unal.edu.co; 3Faculdade de Farmácia, Departamento do Medicamento, Universidade Federal da Bahia, Salvador 40110-909, Bahia, Brazil; izabel.alves@ufba.br; 4Programa de Pós-Graduação em Farmácia, Universidade Estadual da Bahia, Salvador 40110-909, Bahia, Brazil

**Keywords:** snake venom, innovations, pharmacological applications, snake venom-derived therapeutics, patents

## Abstract

Active compounds from natural sources, particularly snake venoms, are crucial for pharmaceutical development despite challenges in drug discovery. Snake venoms, historically used for medicinal purposes, contain bioactive peptides and enzymes that show therapeutic potential for conditions such as arthritis, asthma, cancer, chronic pain, infections and cardiovascular diseases. The objective of this study was to examine pharmacological and biomedical innovations by identifying the key research trends, the most studied snake species, and their therapeutic applications. A systematic review of patents related to snake venoms was conducted using the European Patent Office database, Espacenet, covering 2014 to mid-2024. The search employed the keyword “venom,” applying IPC classification A61K38/00, resulting in 31 patents after screening. A PubMed survey on “snake venom derivatives innovations” was conducted to compare the scientific literature volume with the identified patents. This review highlights the therapeutic potential of snake venom-derived products for coagulation disorders, cancer, inflammation, and pain management. Despite challenges in pharmacokinetics and venom variability, advancements in biotechnology offer promise for personalized therapies. The future of snake venom-based treatments appears promising for addressing complex medical conditions.

## 1. Introduction

Active compounds derived from natural sources are invaluable in the development of pharmaceuticals [1]. Despite the challenges in drug discovery, natural products isolated from medicinal plants and other sources remain crucial in the quest for novel medications [1]. Recently, animal venoms have emerged as a significant source of potential therapeutic agents. Drugs derived from snake venom have demonstrated efficacy across a range of medical conditions [2]. For centuries snake venoms, bile, oils, meat, fat and body parts have been used across the world to treat diverse illnesses [3]. Ancient cultures from Asia, Africa, Europe and the Americas have described their benefits. Although snake venom is inherently toxic, it contains several peptides and enzymes with significant biological activities [1]. Among the species that have been recognized for their medical properties are cobras, vipers, pit-vipers and pythons [3]. Their derivatives have been used to alleviate arthritis, asthma, eczema, pain, inflammation, allergy, migraine, rheumatism, diabetes, skin disorders, cancer, heart conditions and infections [3]. Research on venom components is essential for developing targeted therapeutic strategies due to their specificity and potency [4,5].

Snake venom consists of a complex mixture of peptides, proteins, enzymes, and small molecules, each exhibiting diverse pharmacological properties [6]. Some of the research toxins present in snake venom are three-finger toxins (3FTxs), phospholipases A2 (PLA2s), snake venom metalloproteinases (SVMPs), snake venom serine proteinases (SVSPs), cysteine-rich secretory proteins (CRISPs), L-amino acid oxidases (LAAOs), and C-type lectin-like proteins (CTLPs) [7]. The activity of these toxins can be grouped as follows: myotoxic, neurotoxic, cardiotoxic, anticoagulation, hemostatic disturbances and inhibition of platelet aggregation and antimicrobial [8]. Given the wide range of activities, several bioactive venom peptides and proteins from untapped sources could potentially lead to new therapeutic agents [8].

The research, prospecting, and commercialization of snake venom-derived drugs have gained increasing relevance in the search for new bioactive compounds [7]. This field offers promising potential for the development of innovative therapies, expanding the frontiers of medicine and providing new therapeutic alternatives for various diseases [9]. Characterizing toxins in snake venoms is essential for advancing therapeutic applications, particularly with recent technological advancements in transcriptomics and proteomics that have significantly enhanced this process [10]. These innovations enable the rapid discovery of nearly the entire set of toxins in snake venom, a feat previously unattainable using older methods like reversed-phase high-performance liquid chromatography (RP-HPLC) and mass spectrometry [11]. Knowing the specific activity of snake toxins offers a solid foundation for selecting drug candidates, enabling researchers to move beyond traditional high-throughput screening methods [10]. These tools are helping to speed up the identification of potential drug candidates [11].

Innovating and exploring new compounds derived from snake venoms are critical for addressing unmet medical needs and expanding therapeutic options. Snake venoms offer a diverse range of bioactive molecules with specific mechanisms, presenting potential breakthroughs for treating conditions such as chronic pain, cardiovascular diseases, and metabolic disorders. As conventional drug discovery encounters limitations in chemical diversity, the distinctive properties of snake venom toxins represent an untapped reservoir for novel therapeutics. Continued research into the molecular structures of these toxins will facilitate the development of targeted therapies, enhancing safety and efficacy [12].

While unmodified toxins present challenges in administration, stability, and large-scale production, toxinomimetic approaches (modifying toxin structures) have already led to the development of successful drugs [10]. Emphasizing innovative strategies in this field will not only enhance our understanding of venom biology but also drive the pharmaceutical industry toward more effective and diverse therapeutic options. Several drugs and patents have emerged in this field, with the FDA approving snake venom-derived medications such as Tirofiban and Eptifibatide, derived from disintegrins found in *Echis carinatus* and *Sistrurus miliarius barbouri*, respectively [12]. Additionally, Captopril, an antihypertensive drug, was developed from a bradykinin-potentiating peptide found in *Bothrops jararaca* [12]. Despite these advancements, there remains a significant gap in the literature regarding the full potential of snake venoms as a source of bioactive peptides for drug development. To address this gap, the objective of this study was to conduct a systematic review of patents related to the use of snake venoms, focusing on pharmacological and biomedical innovations. This analysis aims to identify key research trends, the most studied snake species, and their therapeutic applications, highlighting the need for continued exploration and development in this area.

## 2. Results

This systematic review examines patents related to innovations derived from snake venom, analyzing a total of 31 patents from various countries: China (*n* = 21), The United States (*n* = 4), Republic of Korea (*n* = 3), New Zealand (*n* = 1), and the World Intellectual Property Organization (WIPO) (*n* = 2) (see Figure 1). The patents were registered by a range of institutions, including research institutes (*n* = 3), universities (*n* = 10), and the pharmaceutical industry (*n* = 18) (see Figure 1). This distribution highlights the pharmaceutical industry’s prominent role in patent registrations, especially in China, while also recognizing the contributions of research institutions and universities in different regions.

Regarding the years of patent applications and scientific publications for snake venom-derived innovations between 2015 and 2024, it is observed that publications exceed patent applications in most years, with a significant peak in 2020, where publications reach their highest value, close to 15. Patent applications show a more uniform distribution over the years, with smaller peaks in 2017 and a slight increase in 2024. In 2021 and 2022, both categories maintain relatively high frequencies. The years 2016 and 2023 show lower values in both categories (see Figure 2).

In Table 1 the information is presented as follows: Patent Code, Institution, Year/Country, Species, Innovation Name, Activity and Mechanism, Dose, Assay, and Formulation of the 31 patents. In Table 2 the sequences of amino acids of the innovations described in the patents.

## 3. Discussion

### 3.1. Geographic Distribution of Snake Venom-Related Patents

China has emerged as a leader in the research and patenting of snake venom-derived products. The extensive use of snake venom in traditional Chinese medicine has facilitated significant advancements in this area [14]. China’s long-standing tradition of snake venom use for various ailments has spurred research and development of products with diverse applications. The Global Innovation Index 2023 highlights China as a top innovator among upper–middle-income economies, alongside the Republic of Republic of Korea [15]. This aligns with the high number of patent registrations, particularly in China (*n* = 21). Both China and the Republic of Republic of Korea are ranked among the top three innovation economies in Southeast Asia, East Asia, and Oceania [15]. Furthermore, China, Japan, and the Republic of Republic of Korea are prominent international patent filers, reflecting their increased investment in research and development [16]. The United States also ranks among the top three innovation economies in North America, being in second place in the top ten countries in snake venom publication at a worldwide level from 1933 to 2022 [17]. It stands out with 548 pieces of research related to snake venom characterization, products and innovations [17]. Even though The United States has an extensive trajectory in snake venom research, China leads in research specifically focused on snake venoms; this could be associated to its traditions and perception of snakes in their culture [18].

### 3.2. Ten-Year Range Patent Distribution

Given that the search was carried out from 2014 to 2024, the growing number of patent registrations in the last ten years is evident. The data indicate a gradual increase in applications over the years, peaking in both 2018 and 2019. Notably, there is a decline in applications in the subsequent years, decreasing during the pandemic, with fluctuations observed in 2020 through 2024. This trend reflects the evolving interest and activity in the field of snake venom research and its potential applications. This coincides with a rise in the number of publications related to snake venoms. In 2022, a total of 2999 publications were reported in snake venom-related research, which has gradually increased since the 1960s, with the highest number of documents published in 2020 and a continued increase to date with >15,000 studies published on this topic [17]. In a search carried out in PubMed on the same time (2014–2024) frame and using the same key words, a total of 94 published articles about snake venom pharmaceutical derivatives was found. Figure 2 shows trends in patent applications and publications related to snake venom innovations from 2015 to 2024. The peak occurs in 2020, with a significant rise in publications, while patent applications remain lower. Publications consistently outnumber applications in most years, but 2021 and 2022 show similar patterns in both categories. After a decline in 2023, 2024 indicates a resurgence, particularly in publications, suggesting renewed interest or ongoing developments in the field. This pattern highlights the continued scientific exploration of the biomedical applications of snake venom, though patent applications appear to lag behind, indicating challenges in transforming scientific discoveries into marketable products [17]. The overall trend emphasizes the critical need to bridge the gap between scientific discovery and patentable innovation in this domain [17].

Snakes are the most studied venomous animals, primarily due to their ability to produce larger volumes of venom compared to smaller organisms like scorpions and cone snails, as well as their significant medical importance, especially in tropical countries, where snakebites pose a major public health threat. While these smaller animal venoms also hold therapeutic potential, the abundance of snake venom offers unique research opportunities. Research on snake venoms has evolved significantly, utilizing a diverse range of methodologies that continue to expand [10]. Recent advancements in mass spectrometry and sequencing technologies have enabled comprehensive characterization of venoms as complex systems, allowing for the identification of new toxins and the examination of their effects on prey metabolism post-envenomation [19]. The advent of omics technologies—such as proteomics, peptidomics, transcriptomics, genomics, and metabolomics—has revolutionized venom research, ushering in an era of big data analysis. These methods facilitate the study of venoms at unprecedented levels of detail, enhancing our understanding of their biological roles and therapeutic potential [19]. The distribution of patent registrations over the ten-year period covered by this review vs. publications is illustrated in Figure 2.

### 3.3. Most Used Snake Species in Pharmacological Innovations

Research on snake venoms and their evolution has increased over the past few decades [17]. It predominantly focuses on two large families: *Elapidae* (cobras, kraits, mambas, Australian venomous terrestrial snakes, coral snakes, and sea snakes) and *Viperidae* (true vipers and pit vipers), each exhibiting considerable variability in their venom composition [20]. The function of snake venom is to facilitate predation and act as a defense against predators. Given that snake venom is an adaptative trait, the evolutionary dynamic between snakes and their prey resembles an arms race. Prey develop resistance to toxins, prompting snakes to continuously adapt and optimize their venom composition. Dietary habits are significant drivers of venom evolution, leading to both interspecific and intraspecific variations. For instance, geographic variation in viper venom is often linked to dietary habits or topographical features [21].

There are various trophic adaptation examples, such as the Malayan pit viper’s venom variation corresponding to local prey availability and the Mangrove catsnake’s specialization for birds and lizards. Conversely, the Mojave rattlesnake demonstrates that environmental factors may also play a crucial role in venom composition [2,22]. Another example is the Northern Pacific rattlesnake, which exhibits a complex venom profile influenced by coevolution with prey, genetic distance, and habitat gradients. This complexity suggests that multiple ecological factors shape venom diversity [23]. This biochemical variation can occur both between closely related species and within species. Intra-genus or intraspecific variability in venom composition presents numerous opportunities for pharmaceutical research [9].

This review identified the most frequently studied snake species for venom extraction used in innovations. As shown in Table 3, the wide range of species utilized spans from elapids to vipers (Figure 3). The distinct venom compositions of these species offer numerous research and development opportunities [17]. These venoms can be categorized based on their primary toxic effects: neurotoxic, cytotoxic and cardiotoxic (elapids), hemotoxic and cytotoxic (vipers) [2].

Among elapids, the Chinese cobra (*Naja atra*) stands out with the highest number of patents (*n* = 14) and medical applications, reflecting its extensive study in China. It is not surprising that the most studied snake is the Chinese cobra, as this coincides with the increase in research and investment that the Chinese government has made in technology and innovation. Likewise, China is known for its long tradition of using animal and plant derivatives in its traditional Chinese medicine practices. In addition, it can be seen how China has made efforts to research the scientific foundations of its traditional practices [14].

Similarly, other cobras from Asia and Africa are included in the group of most frequently used snakes in pharmacological innovations. These snakes belong to the group of elapids that include kraits and sea snakes. The venom of elapids is mainly characterized by its neurotoxicity which coincides with the medical applications of these patents, which mainly include pain management [13]. However, in this review, we found that the venom of elapids was also used for other activities such as antitumor [13]; anti-inflammatory [13]; anticoagulant [13]; thrombolytic [13]; antiviral [13]; antibacterial [13] and immunogenic [13]; these indications could be associated with some toxins present in the venom of elapids.

Likewise, vipers and pit vipers from Asia, Europe and the Americas have also been subjects of significant research for their medicinal properties. Its venom is cytotoxic and hemotoxic, these being its main activities. Although most of the indications of the registered innovations can be categorized within the intrinsic activities of venom. Other activities found that were not traditionally associated with the venom of these families, which is expected due to their adaptive processes mainly due to environmental factors. Patents have been registered for the following activities: hemostatic [13], thrombolytic [13], antiplatelet [13], cytotoxic [13], cytolytic [13], anti-inflammatory [13], and antibacterial [13].

The diversity of species studied has led to multiple innovative developments addressing various needs. Registered innovations can be categorized into the following types: formulations (*n* = 14) for treating various medical conditions, synthesized molecules (*n* = 7), preparation methods (*n* = 4), extraction methods (*n* = 3), purification methods (*n* = 2), and manufacturing methods (*n* = 1). These patents have contributed to research on ophidian fauna from different countries, facilitating the exploration of new uses and applications for snake venom-derived products [17].

**Table 3 toxins-17-00136-t003:** Snake venom families, scientific names, common names, and pharmacological activity reported in the literature.

Family	Scientific Name	English Common Name	Pharmacological Activity Reported in Literature	Reference
*Pit Viper*	*Agkistrodon piscivorus piscivorus*	Northern Cottonmouth	Antithrombotic	[24]
	*Bothrops atrox*	Fer-de-Lance, common lancehead	Anticoagulant	[25]
	*Crotalus adamanteus*	Eastern Diamondback Rattlesnake	Anticancer	[26]
	*Crotalus durissus terrificus*	South American rattlesnake	Anticancer, antimicrobial	[27]
	*Gloydius intermedius*	Central Asian pitviper	Anticoagulant and antiplatelet	[28]
	*Protobothrops mucrosquamatus*	Brown spotted pitviper	Anticoagulant and antiplatelet	[29]
	*Trimeresurus fasciatus*	Banded Pit Viper	Anticoagulant	[30]
	*Trimeresurus stejnegeri*	Chinese Green Tree Viper, Stejneger’s Bamboo pitviper	Virucidal activity	[31]
*Viper*	*Cerastes* spp.	–	Antiplatelet activity	[32]
	*Daboia russelii*	Russel’s Viper	Anticoagulant and antiplatelet	[33]
	*Deinagkistrodon acutus*	Chinese Moccasin, Hundred-pace viper	Antithrombotic, Antiplatelet	[34]
	*Gloydius brevicaudus*	Short-tailed Mamushi	Anticoagulant and antiplatelet	[28]
	*Gloydius ussuriensis*	Ussuri Mamushi	Anticoagulant and antiplatelet	[28]
*Elapid—Krait*	*Bungarus fasciatus*	Banded Krait	Virucidal activity and Antibacterial	[25,31]
	*Bungarus multicinctus*	Many-banded Krait	Immunogenic and anti-inflammatory	[35]
*Elapid—Cobra*	*Naja atra*	Chinese Cobra	Anticoagulant	[36]
	*Naja kaouthia*	Monocled Cobra, Monocellate Cobra	Antiviral, neuromodulatory and analgesic activities	[25]
	*Naja melanoleuca*	Central African Forest Cobra, Black and White Cobra	Antimicrobial and Antiviral	[37]
	*Naja naja*	Common cobra, Spectacled cobra	Virucidal activity	[31]
	*Ophiophagus hannah*	King Cobra	Analgesic	[25]
*Elapid—Sea snake*	*Hydrophis cyanocinctus*	Annulated Sea Snake, Dusky-chinned giant sea snake	Antimicrobial and Anti-Inflammatory Activity	[38]

This table summarizes the snake venom families, scientific names, common names, and the pharmacological activities reported in the literature for the species mentioned in the patents included in this review. The species listed are those specifically referenced in the selected patents, offering an overview of their therapeutic potential and the pharmacological effects attributed to their venom components as reported in scientific studies.

### 3.4. Therapeutic Indications, Applications, and Compounds Reported in Snake Venom Innovations

Ancient civilizations have long recognized the medicinal properties of snake venom, attributing its benefits to its toxin mechanisms [39]. Modern science has succeeded in isolating and studying these venoms, leading to numerous biomedical applications [8]. This review found that the primary medical indications for registered innovations include coagulation disorders/thromboembolic diseases, lung cancer, inflammation, chronic obstructive pulmonary disease, antiviral applications, Zika virus infection, hemorrhage, pain, arthritis, gout, rheumatoid arthritis, snake bites, cancer, myasthenia gravis, and hemophilia A; Figure 4. Despite the broad biomedical applications of snake venoms due to their compositional variability, their activities can be classified into several categories: anticoagulant, cytotoxic, anti-inflammatory, antiviral, procoagulant, hemostatic, antinociceptive, cytolytic, thrombolytic, antibacterial, and immunogenic; Figure 5 [40]. The following section will detail the compounds, and their respective pharmacological activities as described in the 31 patents identified in this review.

#### 3.4.1. Anticoagulants and Hemostatic Agents

The coagulation cascade is understood as a series of intricate proteolytic events primarily occurring on the surface of activated platelets [41]. When platelets encounter activated endothelium, they release mediators such as P-selectin and von Willebrand factor, which facilitate microvesicle formation and enhance platelet adhesion at the injury site [42]. These microvesicles then fuse with activated platelet membranes, delivering crucial components like tissue factor and factor VIIa, initiating the coagulation cascade [41].

As clotting factors bind to specific receptors on the platelet membrane, they trigger a series of proteolytic cleavages of zymogens into active enzymes, culminating in thrombin generation. Thrombin is essential for converting fibrinogen into fibrin, leading to the formation of a stable blood clot. This process occurs within the protective environment of the platelet membrane, effectively localizing clot formation to the site of injury while shielding it from circulating anticoagulants [42].

New anticoagulants, especially those derived from snake venoms, are crucial due to their unique mechanisms in targeting blood coagulation pathways. These agents offer potential advantages in treating thrombotic disorders, reducing side effects, and overcoming resistance associated with conventional therapies. Certain enzymes found in snake venoms exhibit the capacity to interact with fibrinogen, which has considerable therapeutic relevance, especially concerning hemostasis. These enzymes can effectively deplete fibrinogen from the bloodstream without converting it to fibrin or causing platelet aggregation. Snake venom proteinases that target fibrinogen can be classified into three main categories: thrombin-like enzymes (thrombin proteases), fibrinogenolytic enzymes, and enzymes that activate plasminogen [12]. Each of these enzyme types disrupts normal coagulation mechanisms, underscoring the potential of snake venoms as a source of novel therapeutic agents for managing hemostatic disorders [12].

#### 3.4.2. Metalloproteinases

One of the larger family of enzymes found in snake venom from vipers principally but also could be present in elapid venom are SVMPs. Snake venom metalloproteinases are grouped into three categories: P-I (metalloproteinase [M] domain only), P-II (M domain and disintegrin-like domain), and P-III (M domain, disintegrin-like domain, and cysteine-rich domain). These enzymes have a cytotoxic and hemotoxic effect, inducing blood coagulation to mediate through fibrin formation, Factor V activation, prothrombin activation, actin dissolvement, or platelet aggregation [43]. Likewise, they also have anticoagulant effects through fibrinolysis, fibrinolytic enzyme activation, or protein C activation. Metalloproteinases hydrolyze extracellular matrix components, leading to the rupture of capillaries and local and systemic bleeding, which manifest as edema, inflammation and myonecrosis. The pharmacological use of metalloproteinases is wide, and the effects of these toxins can be categorized as anticoagulant, clotting factor-activating, or platelet-aggregation [23].

A formulation featuring a PIII-type metalloproteinase from *Naja atra* venom (CN105567666 (A)) [13] was reported that to act as an anticoagulant by hydrolyzing the fibrinogen α chain. This enzyme’s anticoagulant activity is inhibited by metal chelators and reducing agents. The formulation was tested at dosages of 0.3 to 3.0 mg/kg, and its efficacy was demonstrated through in vitro assays in rabbits and rats. Similarly, another formulation from *Naja atra* venom (CN108273067 (A)) [13] also employs a PIII-type metalloproteinase for thrombolytic activity, showing effectiveness in dissolving fibrin deposits. Considering the high prevalence of coagulation disorders around the world. Recent studies have emphasized the potential of metalloproteinases present in snake venoms in managing thrombotic conditions due to their ability to directly target fibrinogen and thrombus formation, contributing to novel therapeutic approaches in hemostasis and thrombosis [44,45].

#### 3.4.3. Fibrinolytic Enzymes

Patent (CN110724678 (A)) [13] relates to an invention involving the development of plasmin derived from viper venom *Agkistrodon acutus* (viper venom plasmin), which is a fibrinolytic enzyme, meaning it functions in the dissolution of blood clots. It was tested at doses of 25, 50, and 100 μg/kg of venom and it demonstrates promising results for managing thrombotic diseases. Fibrinolytic enzymes from snake venoms are gaining attention for their specificity in breaking down fibrin networks in thrombi, providing an alternative to traditional thrombolytics and enhancing treatment outcomes in thrombotic disorders. This medical use is directly associated with hemotoxicity, the primary venom activity of vipers [46,47].

#### 3.4.4. Coagulation Factor Activators

Among the detection technologies of the innovations, a novel method was identified for extracting and purifying coagulation factor X activators from various snake venoms (CN109943554 (A)) [13]. It enhances hemostatic processes, accelerating thrombin generation at vascular injury sites. Coagulation factor activators derived from snake venoms were shown to significantly improve hemostasis by accelerating clot formation and stabilizing thrombi, making them valuable tools in managing severe bleeding and hemorrhagic conditions. Throughout history, several snake venom-derivative substances can be found to manage hemorrhage, and some approved drugs are available nowadays [40,48].

### 3.5. Anti-Inflammatory Agents

Inflammation is a complex physiological response that occurs in response to tissue damage, which can arise from various sources, including physical injury, ischemic damage due to inadequate blood supply, infections, exposure to toxins, or other forms of trauma. This multifaceted process involves a series of cellular and molecular events aimed at repairing damaged tissue and promoting cellular proliferation at the injury site. Key players in inflammation include immune cells such as macrophages, neutrophils, and lymphocytes, which are recruited to the site of injury and induce the healing process through the release of pro-inflammatory cytokines and growth factors [49].

The inflammatory response is characterized by several hallmark features, including redness, heat, swelling, and pain, which result from increased blood flow and vascular permeability. While acute inflammation is essential for initiating repair and restoring homeostasis, it can become chronic if the underlying cause persists or if regulatory mechanisms fail to terminate the response. Chronic inflammation is associated with various pathological conditions, including autoimmune diseases, chronic infections, and cancer. Understanding the intricate physiology of inflammation not only provides insights into normal healing processes but also highlights potential therapeutic targets for managing inflammatory diseases [49,50].

As well as in the patents found in this study, recent research has highlighted the potential of snake venom as a promising source of novel anti-inflammatory agents. For instance, recent research has identified specific venom components that can modulate inflammatory pathways, offering promising therapeutic applications for chronic inflammatory conditions. It was demonstrated that certain peptides from snake venom possess significant anti-inflammatory effects by inhibiting key cytokines involved in the inflammatory response [7]. Additionally, it was reported that venom-derived compounds can effectively reduce inflammation in animal models, suggesting their potential as alternative treatments for inflammatory diseases. As the understanding of venom biology deepens, the development of these natural products into clinically relevant therapies may provide innovative approaches to managing inflammation and its associated disorders [51].

Among the innovation assay, a new anti-inflammatory peptide named DAvp-1 was found, obtained from *Deinagkistrodon acutus* venom (CN113388020 (A)) [13]. This peptide antagonizes tumor necrosis factor-alpha (TNF-α) and exhibits significant anti-inflammatory effects in ulcerative colitis models. The formulation was prepared at a concentration of 200 μg/mL and dosed at 500 μg/kg for therapeutic evaluation. Anti-inflammatory peptides from snake venoms, such as DAvp-1, target key inflammatory pathways and cytokines, offering potential for novel treatments in inflammatory diseases and autoimmunity [52].

A formulation derived from heat-denatured *Naja atra* venom (CN104434981 (A)) [13] shows notable anti-inflammatory properties against acute lung inflammation. Similarly, another anti-inflammatory formulation from *Naja atra* venom (KR20190102909 (A)) [13] features a low-molecular-weight peptide and demonstrates effective anti-inflammatory activity. Research into snake venom-derived anti-inflammatory agents highlights their ability to modulate immune responses and reduce inflammation, providing alternative therapeutic options for chronic inflammatory conditions [53,54].

### 3.6. Cancer Therapies

Given the complexity of cancer and the various challenges posed by current therapies, the need for new therapeutic approaches is critical. Existing treatments often have limitations in efficacy, particularly due to the complexity and diversity of cancer types. The development of the disease is a multistep process, characterized by multiple genetic alterations that provide growth advantages, leading to the transformation of normal cells into malignant ones. Given this complexity, novel medications are urgently needed to target these mechanisms more effectively, minimize side effects, and improve survival rates for patients with various types of cancer [49,55].

This malignant transformation is marked by several hallmark characteristics: self-sufficiency in growth signals, insensitivity to antigrowth signals, evasion of apoptosis, unchecked proliferative potential, enhanced angiogenesis, and the ability to metastasize. Each of these changes involves intricate interactions among various signaling pathways, contributing to the cancerous phenotype. Furthermore, emerging evidence suggests that inflammation plays a significant role in the promotion and progression of cancer, potentially facilitating the emergence of these malignant traits. Understanding the complex physiology of cancer is crucial for developing targeted therapies and improving clinical outcomes [49,56].

The potential of snake venom as a source of innovative cancer therapy has earned significant attention in recent years. New anticancer therapies derived from snake venoms are vital due to their potent bioactive compounds, such as disintegrins and phospholipases, which target cancer cells with high specificity. These agents offer promising alternatives for overcoming drug resistance, minimizing side effects, and enhancing treatment efficacy. However, snake venoms contain a rich diversity of bioactive compounds, that exhibit cytotoxic and anti-proliferative properties [57]. Recent studies [57,58] have shown that specific components of snake venom can selectively target cancer cells while sparing normal tissues. Several peptides from snake venom were identified that induce apoptosis in various cancer cell lines by activating intrinsic apoptotic pathways [58]. Furthermore, certain venom-derived compounds can inhibit tumor growth and metastasis in animal models, highlighting their therapeutic potential. These findings underscore the importance of exploring snake venom as a novel source of anticancer agents, paving the way for the development of new strategies in cancer treatment [59].

### 3.7. Cytotoxins and Polypeptides

Cytotoxins and polypeptides from snake venoms, such as those from *Naja atra* (CN107737333 (A)) [13] and (CN107929717 (A)) [13], show efficacy in cancer treatment by inducing apoptosis and cellular damage in tumor cells. The formulation with these cytoxins and polypeptides was tested in various cancer cell lines and murine models. Snake venom-derived cytotoxins are being explored for their potential to selectively target and destroy cancer cells, offering innovative approaches to cancer therapy and highlighting their role in precision medicine [60,61,62].

### 3.8. Fusion Proteins

In one of the patents identified in this work, a novel anticancer treatment stands out for its originality, combining a single-chain anti-IL-4R antibody with L-amino acid oxidase enzyme derived from *Trimeresurus fasciatus* (CN106632686 (A)) [13]. It targets lung cancer cells, demonstrating significant cytotoxic effects and apoptosis induction. Fusion proteins combining monoclonal antibodies with venom-derived enzymes represent a cutting-edge approach in targeted cancer therapy, leveraging the specificity of antibodies and the potency of venom toxins to enhance therapeutic efficacy [63]. This offers a new approach to cancer treatment by reducing its adverse effects.

### 3.9. Pain Management

Pain is defined as an unpleasant sensory and emotional experience associated with actual or potential tissue damage. Pain often serves as a symptom indicating tissue injury, which is linked to nociceptive pain arising from the activation of nociceptors—pain receptors found in the skin, muscles, joints, and organs. These nociceptors are stimulated by extreme temperatures, significant pressure, and chemical signals released during inflammatory processes or cell death. The transmission of these signals conveys critical information about the location, type of injury, and nature of the pain (acute or chronic), prompting individuals to react appropriately, such as moving away from the source of pain or seeking medical help [64,65].

The biological function of pain is essential for tissue repair, as it encourages immobilization of the affected area, allowing for rest and recovery while promoting the search for medical assistance. Pain is often accompanied by inflammatory processes, leading to the use of anti-inflammatory medications in pain management. Understanding the physiology and definition of pain is vital for developing effective treatments and addressing the complex mechanisms involved in pain perception and response [66,67].

The exploration of snake venom as a source of novel pain management agents has gained momentum in recent years, driven by the need for effective alternatives to conventional analgesics. Snake venoms are composed of a complex mixture of bioactive peptides and proteins that have evolved to modulate pain pathways in their prey. Recent research has identified specific venom components that exhibit potent analgesic properties, demonstrating the potential to alleviate pain through various mechanisms [68]. These components often interact with receptors involved in pain signaling, such as ion channels and G-protein-coupled receptors, leading to reduced nociceptive responses. Additionally, studies have highlighted the ability of certain venom-derived peptides to inhibit inflammation, further contributing to their analgesic effects [69]. The ongoing investigation into the molecular mechanisms of these venom components holds promise for the development of new, targeted therapies for pain management, potentially offering safer and more effective options for patients.

### 3.10. Antinociceptive Agents

Some compounds and formulations derived from snake venoms, which offer novel approaches to pain management, were identified in this study. These include a crotoxin-based from *Crotalus durissus terrificus* (US2015110770 (A1)) [13] and an antinociceptive from cobra venoms (US201933657 (A1)) [13]. Both exhibit significant antinociceptive properties in pain models such as postoperative pain in Rats, Von Frey test, formalin, hot-plate and acetic acid writhing tests. Recent advancements in venom-derived antinociceptives emphasize their ability to target pain pathways with high specificity, providing effective relief in various pain models and suggesting potential clinical applications in pain management. Additionally, it had been reported that these new mechanisms avoid the opioid pain pathways, helping with the abstinence syndrome [70,71,72].

### 3.11. Neurotoxins

Neurotoxins are a major compound in elapids venom. From *Naja atra* (CN114409757 (A)) [13], a neurotoxin formulation has been demonstrated to effectively block nerve impulse transmission at neuromuscular junctions, offering significant pain relief. Neurotoxins from snake venoms are increasingly recognized for their potential to treat chronic pain conditions by interfering with neurotransmission and modulating pain pathways, representing a promising area of research in analgesia [68,73].

### 3.12. Antimicrobial Agents

The global issue of antibiotic resistance has intensified the search for novel antimicrobial agents, leading researchers to explore unconventional sources, that include animal venoms. Among these, snake venom has gained significant interest due to its complex mixture of bioactive compounds. Recent studies have identified various components of snake venom, such as peptides and proteins, that exhibit potent antimicrobial properties against a broad spectrum of pathogens, including bacteria, fungi, and viruses [74].

These bioactive molecules often possess unique mechanisms of action, which can disrupt microbial cell membranes, inhibit protein synthesis, or interfere with essential metabolic pathways. Such diverse functionalities make them promising candidates for the development of new therapeutic agents. Advances in biotechnology and molecular biology have facilitated the characterization and synthesis of these venom-derived compounds, enabling the exploration of their structure–activity relationships and potential applications in clinical settings [54,75].

As antimicrobial resistance continues to pose a global health threat, the exploration of snake venom as a source of new antimicrobial agents represents a promising frontier in the quest for innovative solutions to combat resistant infections. This review found recent innovations in snake venom-derived antimicrobial molecules.

#### 3.12.1. Antibacterial Agents


*Snake Venom Polypeptides*


The development of antibacterial agents with peptides represents a promising therapeutic approach in response to the growing bacterial resistance to conventional antibiotics. Antimicrobial peptides (AMPs) are natural defense molecules, present in many organisms, with the ability to effectively destroy bacteria by targeting cell membranes or inhibiting essential metabolic processes. Moreover, they have a broad spectrum of action, being effective against Gram-positive and Gram-negative bacteria, as well as fungi and viruses. Current challenges include optimizing their stability, toxicity, and bioavailability to develop safe and effective therapies for clinical use [76,77]. Many snake venom peptides have demonstrated their efficacy as antibacterial agents. Most snake venoms are thought to contain some form of antibacterial substance, due to the importance of preventing pathogens present in the prey from infecting the snake during feeding. This review found a study (CN117343131 (A)) [13] that explored a range of snake venom polypeptides for antibacterial applications, showing potential against bacterial infections through various assays. Snake venom polypeptides exhibit antimicrobial properties that can disrupt bacterial cell membranes or inhibit essential bacterial enzymes, offering a novel approach to antibiotic resistance and bacterial infections [78,79,80].

#### 3.12.2. Anti-Viral Agents


*Broad-Spectrum Antivirals*


As mentioned above, compounds from snake venoms are efficient antimicrobials. A formulation derived from the venoms of multiple snake species, including *Naja naja* and *Bungarus* both elapids, (CN111617108 (A)) [13], demonstrated broad-spectrum antiviral activity against several viruses, including influenza A and B, HIV, and hepatitis B virus. Snake venom-derived antiviral agents are being investigated for their ability to interfere with viral replication and entry of the virus to the cell, providing a promising and efficient alternative to conventional antiviral therapies [78,81,82].

### 3.13. Anti-Venom Innovations


*Neutralizing Anti-Venoms*


In 2017, the World Health Organization (WHO) classified snake envenomation as a high-priority neglected disease, highlighting its severe impact: up to 2.7 million poisonous bites annually, resulting in approximately 100,000 deaths and many more disabilities. The primary treatment for snake envenomation involves administering specific antivenoms tailored to the snake species involved, alongside supportive care [83].

Neutralizing anti-venoms, which are immunobiological preparations composed of specific immunoglobulins or immunoglobulin fragments that bind to and inactivate toxins present in venom, represent one of the most traditional and common uses of snake venoms [84]. These formulations are produced by immunizing animals typically horses or sheep with venoms and are crucial for treating envenomations by neutralizing toxic effects and mitigating systemic damage. However, the use of animal-derived antivenoms is not without risks; they can provoke hypersensitivity reactions due to their foreign protein content, leading to potentially life-threatening conditions such as severe anaphylaxis. Additionally, antivenoms are often expensive and their availability is limited in many regions [85,86]. Thus, there is a critical need for innovative approaches to improve the efficacy, safety, and cost-effectiveness of antivenoms, which could significantly enhance treatment outcomes for snakebite victims, offering better treatment outcomes for patients affected by snakebites.

Recent formulations developed from the venoms of various Republic of Korean pit viper species (KR20220170290 (A)) [13] have shown effective antidote properties for neutralizing snakebite effects. Advances in antivenom development are focused on enhancing neutralization efficacy, safety, and selectivity to meet the urgent need for effective treatments for snakebite envenomation. Creating novel formulations is essential for improving therapeutic outcomes by increasing efficacy, minimizing side effects, and addressing challenges such as stability and specificity. By innovating new antivenom agents, we can significantly enhance treatment options for snakebite victims, ultimately reducing morbidity and mortality associated with envenomation [84,87,88].

### 3.14. Extraction and Stabilization Methods

The administration and use of snake venom toxins, as therapeutic agents, often presents significant challenges. While these toxins exhibit considerable resistance to proteolysis due to their numerous disulfide bridges, their oral bioavailability remains low, largely because of difficulties in permeating cell membranes. This is reflected in the limited effectiveness of most snake-venom-derived drugs when administered orally. Instead, these toxins tend to demonstrate efficacy in vivo primarily through parenteral injection, a method that aligns with their evolutionary adaptation for bioactivity when delivered via a snake’s fangs to immobilize prey or dissuade predators [89].

Intravenous administration, although the least invasive route that yields positive results, limits the appeal of these toxins for drug development. Additionally, the high costs and complexities associated with their large-scale synthesis, extraction, purification, and heterologous expression pose further challenges. To address these issues, the ongoing research and development of extraction and stabilization methods for snake venom toxins are vital. Such advancements not only enhance the feasibility of using these bioactive compounds in drug discovery but also pave the way for more accessible and effective therapeutic options [89].

The extraction and stabilization of bioactive molecules derived from snake venom have emerged as critical areas of research, driven by the therapeutic potential of these complex compounds. However, the intricate composition of venom presents challenges in terms of efficient extraction and stabilization, which are essential for the development of reliable pharmaceutical applications [11].

Recent advances in extraction methodologies, such as ultrafiltration, solid-phase extraction, and chromatographic techniques, have improved the yield and purity of venom-derived molecules. These techniques enable researchers to isolate specific components while minimizing the degradation of sensitive bioactive compounds. Moreover, innovative stabilization strategies, including formulation in liposomes, nanoparticles, and hydrogels, have been developed to enhance the stability and bioavailability of these molecules in biological systems [90,91,92].

Despite snake venom’s potential for the development of new therapeutic agents, one of the major barriers includes the stability of its toxins. For example, they have poor oral absorption, chemical instability and a short half-life. Considering the above innovations, targeting these barriers is necessary. In this review, we highlight the following innovation.

In our review, we also found methods for stabilizing snake venom enzymes. Some of the methods include using aspartic acid (CN108079285 (A)) [13] and glutamate (CN108273067 (A)) [13] to enhance enzyme stability and efficacy. Another method involves the immobilization of a coagulation factor activator (CN108743924 (A)) [13]. Considering that one of the barriers to using snake venoms in the development of new drugs is its physical instability, improved stabilization techniques contribute to the development of more reliable and effective venom-based therapeutics, ensuring a longer shelf life and consistent therapeutic outcomes [84,90,91].

The field of snake venom-based therapies is only beginning to reveal its potential. As new technologies facilitate the extraction, stabilization, and modification of these compounds, it is expected that new therapies will advance from the laboratory to the market, transforming the treatment of various diseases. However, challenges remain, such as the standardization of toxins and overcoming regulatory barriers. With continued investment in research and development, the future of these therapies promises to bring innovative solutions to some of today’s most challenging medical problems.

## 4. Conclusions

This review explores the potential of snake venom-derived products for treating a variety of medical conditions, such as coagulation disorders, cancer, inflammation, infections, and pain-related issues. FDA-approved drugs like Captopril and Aggrastat (Tirofiban) have already demonstrated the therapeutic value of snake venom-based compounds. Ongoing clinical trials, such as those for Alfimeprase and Viprinex, are investigating additional compounds for thrombolytic and antithrombotic activities. Other promising candidates, including Cenderitide and Contortrostatin, show potential in treating hypertension and inhibiting platelet aggregation, respectively.

While some compounds, like Cobratide, are already approved in certain regions, many others face challenges in advancing due to gaps in understanding their pharmacokinetics, safety profiles, and regulatory hurdles. The variability in venom composition complicates the standardization of products for clinical use. Despite these obstacles, the unique mechanisms of action found in snake venom derivatives provide opportunities for novel therapies, particularly for complex conditions like cancer and chronic inflammation.

Looking ahead, advancements in biotechnology and the use of “omics” technologies (proteomics, transcriptomics, metabolomics) will likely drive further innovation in this field. These technologies will enable more precise identification of venom components and their interactions with biological systems, accelerating the development of personalized treatments. While the potential is significant, critical challenges remain in clinical data, regulatory approval, and product standardization.

## 5. Materials and Methods

A systematic review of patents was conducted using the official database of the European Patent Office (Espacenet). The search covered a 10-year period from 2014 to 2024, with searches carried out in July 2024. The search strategy included the following criteria: the keyword “venom” in the title, a publication date ranging from 31 December 2014 to 30 June 2024, and the IPC classification A61K38/00 (medicinal preparations containing peptides). The Boolean connector AND was used, with no language restrictions applied.

A total of 97 patents were initially identified for potential inclusion. After removing duplicates, 89 patents were screened based on their titles and abstracts, and only those related to snake venom were included. A total of 56 patents were excluded, including those related to bee venom, scorpion venom, and other substances. The flow of patents reviewed is illustrated in the PRISMA diagram in Figure 6. Additionally, a survey of research articles on “Snake Venom Derivatives Innovations” was carried out using the PubMed literature database to compare the number of related papers with the patents found in the patent databases. This survey was also conducted in July 2024.

We utilized an AI-based tool for reviewing and ensuring the accuracy of the English translation, as well as for proofreading and grammar checking in certain sections of the manuscript. This tool was employed selectively to enhance the clarity and quality of specific parts of the text, ensuring that they met the required linguistic standards.

## Figures and Tables

**Figure 1 toxins-17-00136-f001:**
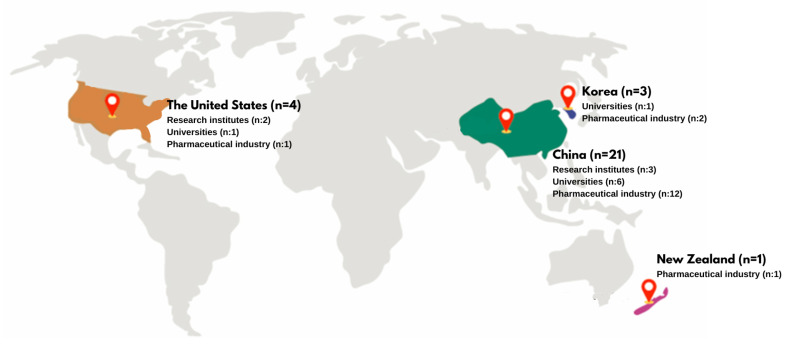
Geographic distribution of snake venom-related patents, highlighting the number of patents registered in each country. This figure illustrates the distribution of patent applications related to snake venom innovations across various institutions and regions.

**Figure 2 toxins-17-00136-f002:**
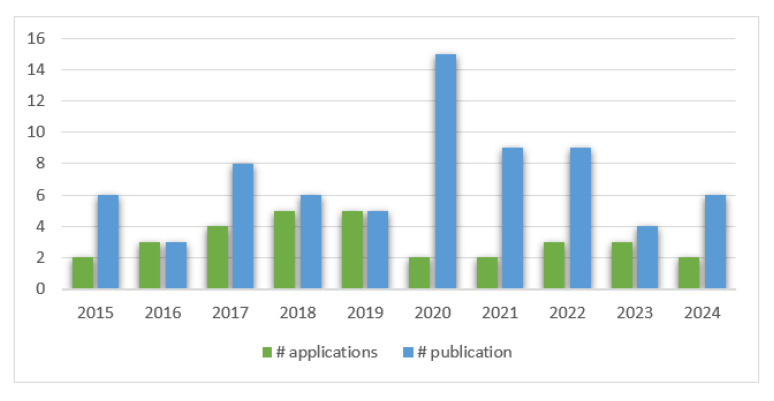
Trends in patent applications vs. publications related to snake venom innovations. This figure presents the annual number of patent applications and publications related to snake venom innovations from 2014 to 2024.

**Figure 3 toxins-17-00136-f003:**
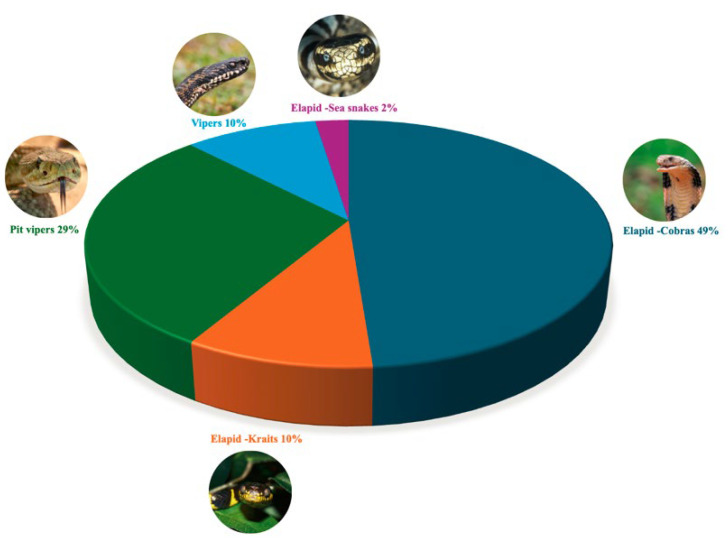
Distribution of bioactive compound sources from various snake families found in innovations in the patents. This graph displays the distribution of bioactive compounds derived from different snake species, highlighting the number of compounds identified in, pit vipers, vipers, and Elapids such as cobras, kraits and sea snakes.

**Figure 4 toxins-17-00136-f004:**
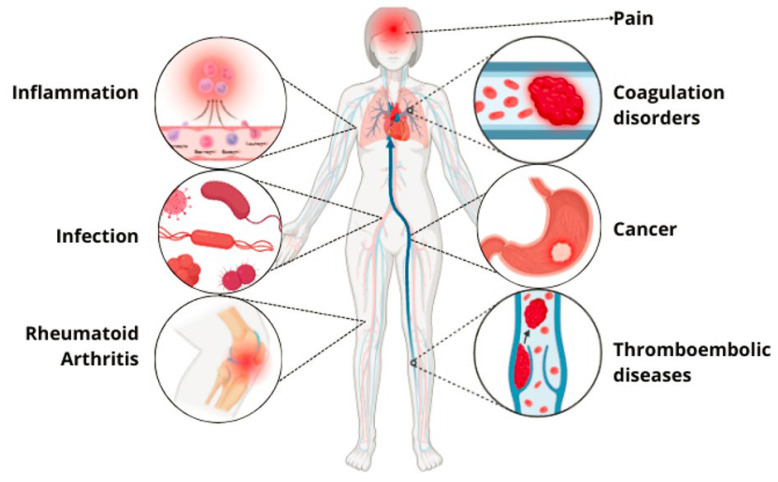
Therapeutic indications of pharmacological innovations derived from snake venom.

**Figure 5 toxins-17-00136-f005:**
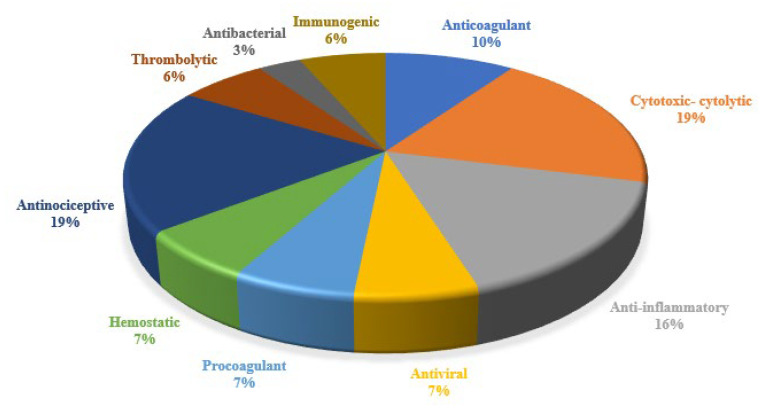
Distribution of pharmacological activities reported in innovations patents.

**Figure 6 toxins-17-00136-f006:**
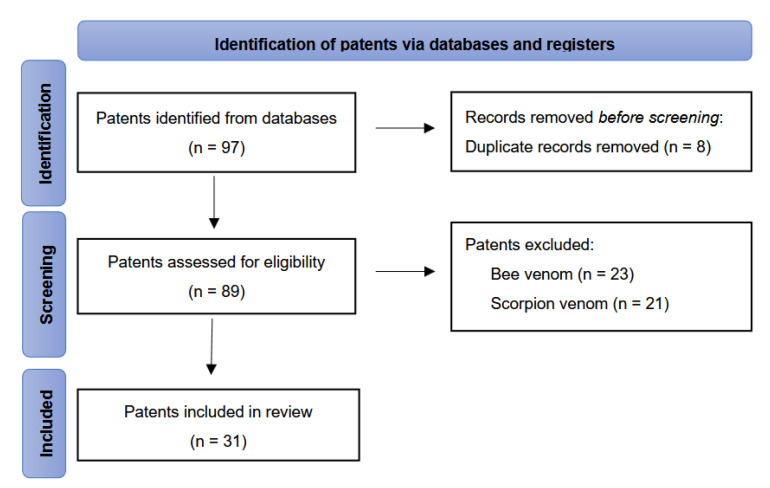
PRISMA flow diagram illustrating study selection and screening.

**Table 1 toxins-17-00136-t001:** Snake venom patents published in European Patent Office (2014–2024).

No	Cod.	Institution	Year/Country	Species	Active Compound	Activity and Mechanism	Administration Route	Dose	Assay
1	CN105567666 (A)	Pharmaceutical industry	2016/China	*Naja atra*	PIII type metalloproteinase	Anticoagulant: The target protein has the activity of hydrolyzing fibrinogen α chain, and this activity can be completely inhibited by metal chelators EDTA, EGTA, 1,10-phenanthroline and reducing agent DTT.	I.V	0.3 and 3.0 mg/kg	**Activity**: In vitro anticoagulation platelet aggregation rate on rabbits Anticoagulant effect in rats
2	CN106632686 (A)	University	2017/China	*Trimeresurus fasciatus*	Anti-IL-4R single-chain antibody and a snake venom L-amino acid oxidase fusion protein.	Citotoxic: Direct killing and apoptosis-inducing effects on a variety of cancer cells, selectivity lung cancer	I.P.	100, 50, 20 mg/kg	**Activity**: MTT assay, effects of fusion protein on human lung adenocarcinoma cell line, effects of anti-IL-4R single-chain antibody and LAAO fusion protein on tumor inhibition rate and life extension rate of mice bearing human lung adenocarcinoma **Safety**: Cell line H460
3	CN113388020 (A)	Research institute	2021/China	*Deinagkistrodon acutus*	Peptide DAvp-1	Anti-inflammatory: Antagonize TNF-α	I.P	500 μg/kg	**Activity**: Ulcerative colitis mouse model.
4	CN104434981 (A)	Pharmaceutical industry	2015/China	*Naja atra*	Physically treated cobra venom	Anti-inflammatory: Inhibit acute lung inflammation induced by lipopolysaccharide	Orally	10–3000 μg/kg animal/1 to 1000 μg/kg per person.	**Activity**: Acute lung injury induced by LPS model
5	CN111617108 (A)	Pharmaceutical industry	2020/China	*Naja naja, King cobra, Bengal cobra, Bungarus, Coral snake, Black mamba*	Elapheidae postsynaptic neurotoxin, cardiotoxin, cytotoxin, phospholipase A2	Anti-viral: Membrane toxins can easily pass through the Membrane structure and then destroy internal structures such as mitochondria and lysosomes, reversibly bind to nicotinic acetylcholine receptors	NA	1.03, 0.39, 0.23, 0.32 and 0.89 (unit: μg/mL)	**Activity**: Virus plaque
6	CN115957303 (A)	University	2023/China	*Hydrophis cyanocinctus*	Polypeptide Hc-CATH	Anti-viral: Destroy the viral envelope, induce the leakage of the viral genome, and thus inactivate viral particles	I.V	5 mg/kg	**Activity**: Zika in vitro/in vivo infection, **Safety**: Cytotoxicity assay in Vero cells
7	WO2023168743 (A1)	Pharmaceutical industry	2023/WIPO	NA	Viper venom hemocoagulase (Slounase)	Procoagulant and hemostatic: Reverse the anticoagulant effect of FXa inhibitors and completely restore thrombosis in human blood.	NA	0.014–0.077 U/mL	**Activity**: In vitro thromboelastography
8	CN108079285 (A)	Pharmaceutical industry	2018/China	*Bothrops moojeni*	L-aminobutanedioic acid stabilizer for snake venom enzyme: Defibrase	Stabilizer: for making snake venom enzyme preparation safer, more effective, more stable.	NA	NA	Stability Test
9	US2019336572 (A1)	Research institute	2019/United states	*Naja tripudians, N. siamensis, N. naja, N. atra, N. kaouthia, and O. hannah*	Cobra venom	Antinociceptive: Target the cholinergic system by blocking the activity of acetylcholine	Orally	0.1 to about 0.5 mg/mL	Stability Test, **Activity**: Formalin, hot-plate and acetic acid writhing tests
10	KR20190102909 (A)	Pharmaceutical industry	2019/Republic of Korea	*Naja atra*	Low molecular peptide isolated from the heat-treated snake venom	Anti-inflammatory: NA	NA	50 μg/mL	**Safety**: Cytotoxicity
11	US2022362356 (A1)	Pharmaceutical industry	2022/United states	*Agkistrodon piscivorus piscivorus or Naja melanoleuca*	Extract of snake venom	Anti-inflammatory: Increase the level of regulatory T cells, increased all of the level of each combination of CD4 positive, CD25 positive, can improve the inflammatory symptoms of rheumatoid arthritis.	Orally/skin application	0.1 to 50 mg/kg	**Activity**: Arthritis model mice
12	KR20220170290 (A)	University	2022/Republic of Korea	Republic of Korea pit viper *(Gloydius brevicaudus, Gloydius intermedius, Gloydius ussuriensis)*	Anti-viper serum/anti-viper viper serum	Anti-venom: venom neutralizing effect	I.M	50 μL/100 μL)	**Safety**: Lethality test
13	US2015110770 (A1)	Research institute	2015/United states	*Crotalus durissus terrificus*	Crotoxin compositions	Cytolytic: NA	I.V	0.0012–0.01 mg/kg	**Activity**: Anti-tumor activity both in vitro and in vivo, **Safety**: LD50
14	CN110724678 (A)	University	2020/China	*Agkistrodon acutus*	Agkistrodon halys venom fibrinolysin- plasmin	Thrombolytic: Remove the fibrin gel blocks deposited in the thrombus on the blood vessel wall, dissolve the thrombus	Intravenous drip	25 μg/kg, 50 μg/kg and 100 μg/kg	**Activity**: Determination of fibrinolytic enzyme activity, thrombolytic assay, Anticoagulant effect in mice
15	CN108273067 (A)	Pharmaceutical industry	2018/China	*Bothrops atrox*	Glutamic acid stabilizer for snake venom enzyme: Defibrase	NA	NA	NA	Accelerated stability test
16	CN108743924 (A)	Pharmaceutical industry	2018/China	NA	Snake venom coagulation factor activator	Procoagulant: clotting factor, blood coagulation X factor activator.	NA	NA	NA
17	CN105497873 (A)	University	2016/China	*Cobra or Agkistrodon*	Light-controlled targeted snake venom polypeptide zinc nanoformulation	Antinociceptive: Central effect.	I.P	2 mg/kg	**Activity**: Hot plate
18	CN109943554 (A)	Pharmaceutical industry	2019/China	*Vipera ruselli, Bothrops, Deinagkistrodon, Bungarus, Cerastes, Calloselasma, Ophiophagus, Crotalus adamanteus and/or Naja*	Coagulation factor X activator from snake venom.	Hemostatic: Factor X activators activate factor X at the site of blood vessel damage, promoting the generation of thrombin.	NA	NA	NA
19	NZ753297 (A)	Pharmaceutical industry	2021/New Zealand	*Agkistrodon acutus*	Recombinant Agkisacutacin	Antiplatelet: Inhibiting platelet aggregation	NA	2 μg	**Activity**: GPIb binding activity and antiplatelet agglutination activity
20	CN109929020 (A)	Pharmaceutical industry	2019/China	*Naja, Naja atra*	Neurotoxin	Antinociceptive: NA	NA	NA	NA
21	KR20190007161 (A)	Pharmaceutical industry	2019/Republic of Korea	*Naja atra, Naja kaouthia*	Modified cobra venom	Anti-inflammatory and antinociceptive: NA	Orally, external and injectable	300 μg/kg	**Activity**: Mouse Ear Swelling Assay **Safety**: LD50
22	CN107737333 (A)	Pharmaceutical industry	2018/China	*Naja atra*	4-aminoquinoline derivative and cytotoxin-CTX1	Anti-tumor: CTXs-mediated cancer cell damage is achieved by destroying lysosomes, induce apoptosis and necrosis of a variety of tumor cells.	NA	NA	**Activity**: Citotoxicity breast cancer cell line MCF7, human acute myeloid leukemia cell line KG1a
23	CN107929717 (A)	Pharmaceutical industry	2018/China	*Naja atra*	Siramesin and snake venom cytotoxin-CTX1	Anti-tumor: Significant synergistic effect on the growth inhibition of the MCF7 tumor line and could effectively induce late apoptosis and necrosis of MCF7 cells.	NA	NA	**Activity**: Citotoxicity cell line MCF7, Changes in reactive oxygen species during cell death
24	CN115594746 (A)	University	2023/China	*Agkistrodon acutus*	C-type lectin-like protein	Anti-platelet and anti-coagulation: Effect by prolonging TT, APTT and PT pathways.	I.V	0.5 μg/g and 1.5 μg/g	**Activity**: Anticoagulant effect on mice, Platelet count, Coagulation function assay, Tail bleeding time determination
25	CN117756907 (A)	Pharmaceutical industry	2024/China	*Naja atra*	Cobra-peptide and Substance A	Antitumor: inhibits the proliferation of various cells, induce early apoptosis.	NA	NA	**Activity**: Human cervical cancer cell line Hela cells, Flow cytometry assay for cell apoptosis **Stability test**
26	CN107098956 (A)	University	2017/China	*Naja atra*	Cytotoxin-4N	Cytotoxic: Activation or cause apoptosis	NA	NA	**Activity**: Effect on the inhibition of HSC-T6 cell proliferation, Effects on apoptosis of HSC-T6 cells
27	WO2017190263 (A1)	Pharmaceutical industry	2017/WIPO	*Naja atra*	C fragment polypeptide	Antinociceptive: NA	I.M	0.5 mg/mL	**Activity**: Postoperative Pain in Rats, Von Frey test
28	CN114409757 (A)	Pharmaceutical industry	2022/China	*Naja*	Cobra venom neurotoxin	Antinociceptive: High affinity for N-type acetylcholine receptors and can block the transmission of nerve impulse signals at the neuromuscular junction	I.M	23.3 μg/kg	**Activity**: Hot plate test
29	CN117343131 (A)	University	2024/China	*Naja atra, Gloydius brevicaudus, Deinagkistrodon, Trimeresurus stejnegeri, Bungarus multicinctus, Bungarus fasciatus, Protobothrops mucrosquamatus.*	Snake venom polypeptide	Antibacterial: NA	Orally/injected	100 mg/kg	**Activity**: MIC E.coli, Acute toxicity test in mice, Subacute toxicity test in mice, Hemolytic test. **Safety**: LD50 Stability test
30	CN105861476 (A)	University	2016/China	*Naja atra*	PIII type metalloproteinase- Atrase A/Atrase B	Thrombolytic: Hydrolyze the α chain of fibrinogen and show anti-platelet aggregation activity	I.V	3.0 mg/kg–0.3 mg/kg	**Activity**: Determination of fibrinolytic function
31	US2017128544 (A1)	University	2017/United states	*Naja naja, Naja kaouthia*	Humanized cobra venom factor	Immunogenic: Induce a humoral or a cell mediated response of the immune system	I.P	250 μg/kg–500 μg/kg	**Activity**: Murine model of age-related macular degeneration, murine model of gastrointestinal ischemia reperfusion injury.

This table summarizes the results of the systematic review of snake venom-related patents published between 2014 and 2024 in the European Patent Office. It includes key details such as the patent number (No), code (Cod.), institution, publication year and country, species, active compound, its activity and mechanism, the administration route, dosage, and assay methods used. **Legends***: I.V: Intravenous, I.P: Intraperitoneal, I.M: Intramuscular, NA: Not Available, LAAO: L-amino acid oxidase, TNF-α: Tumor Necrosis Factor Alpha, TT: thrombin time, FXa: Factor Xa, APTT: Activated Partial Thromboplastin Time, LD50: Lethal Dose 50, PT: Prothrombin Time, CTXs: cytotoxin, MIC—Minimum Inhibitory Concentration. Hc-CATH: Hydrophis cyanocinctus catechin.*

**Table 2 toxins-17-00136-t002:** Sequences of amino acids of the innovations described in the patents.

Application Number [Reference]	Amino Acids Sequence	Activity
CN107737333 (A) [13]	LKCNKLIPIA SKTCPAGKNL CYKMFMMSDLTIPVKRGCID VCPKNSLLVK YVCCNTDRCN	Anti-tumor
CN115957303 (A) [13]	KFFKRLLKSVRRAVKKFRKKPRLIGLSTLL	Anti-viral
CN109929020 (A) [13]	LECHNQQSSQTPTTTGCSGGETNCYKKRWRDHRGYRTERGCGCPSVKNGIEINCCTTDRCNN	Analgesic
CN117756907 (A) [13]	LECHNQQSSQ TPTTTGCSGG ETNCYKKRWRDHRGYRTERG CGCPSVKNGI EINCCTTDRC NN	Antitumor
WO2017190263 (A1) [13]	KDHRGTRIER	Analgesic
CN114409757 (A) [13]	LECHNQQSSQTPTTTGCSGGETNCYKKRWRDHRGYRTERGCGCPSVKDGIEINCCTTDRCNN	Analgesic

This table presents the amino acid sequences of the bioactive compounds described in the reviewed snake venom patents that were publicly available. The sequences included are those specifically detailed in the patents.

## Data Availability

The original contributions presented in this study are included in the article/Appendix A. Further inquiries can be directed to the corresponding author.

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
