# Peer review of "Innovations in Snake Venom-Derived Therapeutics: A Systematic Review of Global Patents and Their Pharmacological Applications"

_toxins, 2025, doi:10.3390/toxins17030136_

Round 1

Reviewer 1 Report

Comments and Suggestions for Authors

An innovative vision of snake venom offers new possibilities. Venoms are more than lethal cocktails; they can also drive the development of pharmaceutical applications. This manuscript describes the less explored side of snake venom. While the therapeutic potential has been analysed in many manuscripts, the intellectual properties and innovations are novel. Although relevant, many aspects need to be revised. Some parts of the manuscript need clarification, and some are not supported by evidence or data. Below, I include my suggestions, which I hope will help improve the quality of the manuscript.

1. Line 9. The therapeutic potential is much wider. Anticancer and antibacterial properties have been extensively explored. Please revise this sentence.

2. Line 10. Please revise the aim of this study.  Use of venom or snake venom-derived components? The use of whole venom is really challenging due to the toxic effects. The use of purified proteins/peptides or venom-based drugs are more realistic.

3. Lines 10-12. This sentence can be rewritten to improve clarity.

4. Please key words different from those used in the title.

5.  The words background, methods and conclusion should not be included in the abstract section. The numeration is not correct as well.

6. The key contribution is long and do not emphasise the contribution of this study.  The authors describe the focus of the review and its structure, instead of highlight how this manuscript fill a gap in the literature. Please also review Toxins guidelines, which stated one or two sentences for key contributions.

7.  Some sentences mut be supported by references. Please review lines 31-32, 41-43, 50-52, 65-66, 66-69, 167-169, 285-288, 311-312.

8. Lines 32-33. Please focus on snake venom. Sentences related to medicinal plants do not add relevant information to the manuscript.

9. Lines 41-43. Please include references for each application to better illustrate the potential.

10. Lines 41-43. Venom components have also antibacterial (DOI: 10.1002/ddr.21456), anticancer (https://doi.org/10.1016/j.cellbi.2006.11.007), antifungal ( doi: 10.3390/toxins12080500), natriuretic, antiviral (doi: 10.1016/j.ijbiomac.2020.07.178) and antiparasitic properties (https://doi.org/10.1016/j.ijbiomac.2023.124745). Please expand the activities.

11. Lines 46-50. The review presented 7 does not presented evidence of the most researched snake venom toxin families. Please provide a reference with this analysis or rewrite this sentence.

12. Line 52. What are the among others?

13. Line 53. Peptides and proteins.

14. Line 56. Where is the analysis showing the increasing relevance?

15. Line 60. What are the recent technological advancements enhancing this process? Please include examples to illustrate this contribution.

16. Line 60-62. This sentence is confusing. What is a unique bioactivity? It is not clear the difference of venom screening to the traditional screening.

17. Lines 62-64. Please include examples of the use of both strategies. It would be interesting if the authors can provide a table with snake venom toxins purified studies with the biological properties and references, as well toxin-derived peptides. There are many phospholipase-derived peptides with antibacterial, antifungal, anticancer, anti-Leishmania effects, and also dual action. One table can enrich this review.

18.  line 67. Unique mechanism or multiple mechanisms? What are they? Many PLA2-derived peptides for example act by membrane disruption of bacteria, cancer or parasite cells. Please enrich the details of the mechanisms.

19. The last paragraph of the introduction can include the gap in the literature and the need of the analysis presented in this manuscript to reinforce its importance.

20. There are important patents involving snake venom excluded from this publication. For example, Brazilian researchers have developed a heterologous fibrin sealant derived from snake venom. This was patented as well: https://patents.google.com/patent/BR102017008027B1/en, doi: 10.1186/s40409-017-0109-8. I am wondering if the search performed by the authors has some limitations. This need to be discussed in the manuscript. The number can be higher than 31.

21. Table. Please revise scientific names. The species name begins with a lower case letter.

22. Tables. Please add a brief description in addition to the titles.

23. Table 2. The first one is a nucleotide sequence, not amino acid sequence. Please standardise the sequences as well, some used one-letter amino acid code, and one used there-letter amino acid code.

24. I am not sure if the results are biased due to the search. Brazil has some patents that were not included in the manuscript.

25. lines 167-169. This is one of the reasons, but the main one is related to the clinical significance. Snakebites are more life-threatening conditions, with huge impact in tropical countries.

26. Line 169. smaller venoms? This is not clear.

27. line 171. What are the evidences for the expansion of the methodologies?

28. Line 187. Where is the supporting data?

29. Table 3 is limited. Many other snake venoms have been studied.

30. The section neutralising antivenoms does not add relevant information to this manuscript.

31. The conclusion is long and not focused on a critical analysis of the data presented.

32. Lines 631-633. The manuscript needs proofreading.

33. The section Mechanisms involved in the indications of snake venom innovations is confusing and needs to be restructured. The mechanisms are not explored. The title needs to be revised to accurately reflect its content. Instead, authors described biological properties of snake venoms. But authors also mix venom enzyme sections with biological properties sections. I recommend describe only the applications and include examples of venom enzymes in each of the sections.

34. The order of figures needs to be revised.  The figure 2 is presented before the figure 1. The figure 1 appears after figure 5. There are two different figures 5. 

35. Please describe and refer the figures in the main text. Some figures are not mentioned in the main text. 

36. Some paragraphs express the same idea and can be combined. There is no reason for a new paragraph when they similar aspects have been described. 

33. Methods. Some paragraphs express the same idea and be combined.

Author Response

1. Summary

We would like to thank you for taking the time to review this manuscript. We greatly appreciate your constructive comments and suggestions. Below, we provide a point-by-point response to your comments. Corresponding revisions and corrections have been made in the re-submitted files and are highlighted in track changes.

2. Point-by-point response to Comments and Suggestions for Authors

Comment 1: Line 9. The therapeutic potential is much wider. Anticancer and antibacterial properties have been extensively explored. Please revise this sentence.

Response 1: Thank you for pointing this out, we have added some more indications in line 9.

Comment 2:  Line 10. Please revise the aim of this study.  Use of venom or snake venom-derived components? The use of whole venom is really challenging due to the toxic effects. The use of purified proteins/peptides or venom-based drugs are more realistic.

Response 2: Thank you for pointing this out, we have added the correction in line 10.

Comment 3: Lines 10-12. This sentence can be rewritten to improve clarity.

Response 3: Thank you for pointing this out, we have added rewrite that sentence in line 10-12.

Comment 4: Please key words different from those used in the title.

Response 4: Thank you for pointing this out, we have improved the key word to be coherent with the title, line 22.

Comment 5: The words background, methods and conclusion should not be included in the abstract section. The numeration is not correct as well.

Response 5: Thank you for pointing this out, we have removed the headings from the abstract. 

Comment 6: The key contribution is long and do not emphasise the contribution of this study.  The authors describe the focus of the review and its structure, instead of highlight how this manuscript fill a gap in the literature. Please also review Toxins guidelines, which stated one or two sentences for key contributions.

Response 6: Thank you for pointing this out, we have rewritten and improved the key contribution.

Comment 7: Some sentences mut be supported by references. Please review lines 31-32, 41-43, 50-52, 65-66, 66-69, 167-169, 285-288, 311-312.

Response 7: Thank you for pointing this out, we have added the missing reference in lines 31-32, 41-43, 50-52, 65-66, 66-69, 167-169, 285-288, 311-312.

Comment 8: Lines 32-33. Please focus on snake venom. Sentences related to medicinal plants do not add relevant information to the manuscript.

Response 8 : Thank you for pointing this out, we have improved the sentence in lines 32-33.

Comments 9: Lines 41-43. Please include references for each application to better illustrate the potential.

Response 9: Thank you for pointing this out, we have added the reference in lines 41-43.

Comment 10: Lines 41-43. Venom components have also antibacterial (DOI: 10.1002/ddr.21456), anticancer (https://doi.org/10.1016/j.cellbi.2006.11.007), antifungal ( doi: 10.3390/toxins12080500), natriuretic, antiviral (doi: 10.1016/j.ijbiomac.2020.07.178) and antiparasitic properties (https://doi.org/10.1016/j.ijbiomac.2023.124745). Please expand the activities.

Response 10: Lines 41-43. Thank you for pointing this out, however, the purpose of this sentence is to mention the therapeutic  conditions for which snake venom has been used, to provide context, as the text will later delve into the activities of the venom responsible for its use in these conditions. Examples such as antibacterial, antifungal, antiviral, and antiparasitic activities are the reasons why snake venom was used to combat infections. I hope this explanation addresses your request.

Comment 11: Lines 46-50. The review presented 7 does not presented evidence of the most researched snake venom toxin families. Please provide a reference with this analysis or rewrite this sentence.

Response 11 : Thank you for pointing this out, we have rewrite the sentence in line 46-50.

Comment 12: Line 52. What are the among others?

Response 12 : Thank you for pointing this out, we have rewrite the sentence in line 52.

Comment 13:  Line 53. Peptides and proteins.

Response 13: Thank you for pointing this out, we have rewrite the sentence in line 53.

Comment 14: Line 56. Where is the analysis showing the increasing relevance?

Response 14 : Thank you for pointing this out, we have added the reference in line 56.

Comments 15: Line 60. What are the recent technological advancements enhancing this process? Please include examples to illustrate this contribution.

Response 15: Thank you for pointing this out, we have rewritten the paragraph to improve its clarity, lines 60-67.

Comments 16. Line 60-62. This sentence is confusing. What is a unique bioactivity? It is not clear the difference of venom screening to the traditional screening.

Response 16: Thank you for pointing this out, we have rewritten the paragraph to improve its clarity, lines 60-67.

Comment 17. Lines 62-64. Please include examples of the use of both strategies. It would be interesting if the authors can provide a table with snake venom toxins purified studies with the biological properties and references, as well toxin-derived peptides. There are many phospholipase-derived peptides with antibacterial, antifungal, anticancer, anti-Leishmania effects, and also dual action. One table can enrich this review.

Response 17: Thank you for pointing this out, we have rewritten the paragraph to improve its clarity, (lines 60-67).

Comment 18:  line 67. Unique mechanism or multiple mechanisms? What are they? Many PLA2-derived peptides for example act by membrane disruption of bacteria, cancer or parasite cells. Please enrich the details of the mechanisms.

Response 18: Thank you for pointing this out, We changed the word 'unique' since we were referring to the specificity of the mechanisms, not the quantity. We hope it is clearer now (line 71).

Comment 19: The last paragraph of the introduction can include the gap in the literature and the need of the analysis presented in this manuscript to reinforce its importance.

Response 19: Thank you for pointing this out, we have rewritten this paragraph to reinforce the importance of the manuscript ( Lines 79-94).

Comment 20. There are important patents involving snake venom excluded from this publication. For example, Brazilian researchers have developed a heterologous fibrin sealant derived from snake venom. This was patented as well: https://patents.google.com/patent/BR102017008027B1/en, doi: 10.1186/s40409-017-0109-8. I am wondering if the search performed by the authors has some limitations. This need to be discussed in the manuscript. The number can be higher than 31.

Response 20: Thank you very much for your comment. In response, we would like to clarify that the patent you suggested does not meet two of the inclusion criteria defined for this review. Specifically, it does not contain the keyword 'venom' in the title, nor does it match the IPC classification, which is why it was not included. Additionally, only patents that met the inclusion criteria established prior to the search were included in this review following the guidelines outlined in the Cochrane Handbook for Systematic Reviews.

Comment  21. Table. Please revise scientific names. The species name begins with a lower case letter.

Response 21: thank you for your comment, we have revised the names in table 1.

Comment 22. Tables. Please add a brief description in addition to the titles.

Response 22: thank you for your comment, we added descriptions for tables 1,2 and 3.

Comment 23. Table 2. The first one is a nucleotide sequence, not amino acid sequence. Please standardise the sequences as well, some used one-letter amino acid code, and one used there-letter amino acid code.

Response 23: Thank you for pointing this out, we have removed the nucleotide sequence and standardized the codes in table 2.

Comment 24. I am not sure if the results are biased due to the search. Brazil has some patents that were not included in the manuscript.

Response 24: Thank you for your comment. In response: as detailed in the methodology, inclusion criteria were established prior to the search, and all patents meeting these criteria were included. To ensure the quality of the review, the guidelines from the Cochrane Handbook for Systematic Reviews were followed.

Comment 25:. lines 167-169. This is one of the reasons, but the main one is related to the clinical significance. Snakebites are more life-threatening conditions, with huge impact in tropical countries.

Response 25: Thank you for pointing this out, we have rewritten this sentence (line 178-181)

Comment 26: Line 169. smaller venoms? This is not clear.

Response 26: Thank you for pointing this out, we added the missing word (line 181).

Comment 27: line 171. What are the evidences for the expansion of the methodologies?

Response 27: Thank you for pointing this out, we added the reference and in the following sentences provide more details.

Comment 28: Line 187. Where is the supporting data?

Response 28: Thank you for pointing this out, we added the reference.

Comment 29: Table 3 is limited. Many other snake venoms have been studied.

Response 29: Thank you for pointing this out, in Table 3, only the snake species mentioned in the patents are included, and for these, we added additional pharmacological activities reported in the literature. This explanation has also been included in the table description.

Comment 30: The section neutralising antivenoms does not add relevant information to this manuscript.

Response 30: Thank you for pointing this out, this section discusses the patents included in this review on innovations in antivenom production and aims to provide context and explain the patents in this area. The section title was somewhat detached, but it has now been adjusted. We hope this makes the presence of this section in the manuscript clearer.

Comment 31: The conclusion is long and not focused on a critical analysis of the data presented.

Response 31: Thank you for pointing this out, we have rewritten the conclusion.

Comment 32: Lines 631-633. The manuscript needs proofreading.

Response 32: Thank you for pointing this out, we have corrected the paragraph (lines 644-646).

Comment 33: The section Mechanisms involved in the indications of snake venom innovations is confusing and needs to be restructured. The mechanisms are not explored. The title needs to be revised to accurately reflect its content. Instead, authors described biological properties of snake venoms. But authors also mix venom enzyme sections with biological properties sections. I recommend describe only the applications and include examples of venom enzymes in each of the sections.

Response 33: Thank you for pointing this out, We have revised the title to make it more precise.

Comment 34: The order of figures needs to be revised.  The figure 2 is presented before the figure 1. The figure 1 appears after figure 5. There are two different figures 5. 

Response 34: Thank you for pointing this out, We have revised the figures.

Comment 35. Please describe and refer the figures in the main text. Some figures are not mentioned in the main text. 

Response 35: Thank you for pointing this out, We have revised this comment.

Comment 36. Methods. Some paragraphs express the same idea and be combined.

Response 37: Thank you for pointing this out, we have rewritten the methods to avoid redundance.

Reviewer 2 Report

Comments and Suggestions for Authors

Comments to the Author

This study provides a comprehensive systematic review of therapeutic derivatives from snake venom, including patents and scientific publications. As a reviewer, I consider it an excellent resource for researchers exploring pharmacologically relevant molecules.

However, it would be beneficial for the authors to elaborate further on the specific characteristics of each peptide or bioactive molecule that contributed to its success. Additionally, the quality of the figures and images of snakes should be improved. Finally, the authors should include access links to the databases used in the study.

Author Response

1. Summary

We would like to thank you for taking the time to review this manuscript. We greatly appreciate your constructive comments and suggestions. Below, we provide a point-by-point response to your comments. Corresponding revisions and corrections have been made in the re-submitted files and are highlighted in track changes.

2. Point-by-point response to Comments and Suggestions for Authors

Comment 1: it would be beneficial for the authors to elaborate further on the specific characteristics of each peptide or bioactive molecule that contributed to its success.

Response 1: Thank you very much for your recommendation. The information described in the review was expanded compared to what is reported in the included patents, providing more details. It is worth noting that some patents do not provide detailed information, and in many cases, researchers must rely on the literature to gather data, which can still be insufficient or lack the necessary depth.

Comment 2: Additionally, the quality of the figures and images of snakes should be improved.

Response 2: Thank you for your comment. The figures 1, 3 and 5 have been changed to improve their resolution; however, they will be sent separately for inclusion later in the process.

Comment 3: Finally, the authors should include access links to the databases used in the study.

Response 3: Thank you for your suggestion. We will include access links to the databases in the supplementary material.
